# Targeting STAT3 and STAT5 in Tumor-Associated Immune Cells to Improve Immunotherapy

**DOI:** 10.3390/cancers11121832

**Published:** 2019-11-21

**Authors:** Grégory Verdeil, Toby Lawrence, Anne-Marie Schmitt-Verhulst, Nathalie Auphan-Anezin

**Affiliations:** 1Laboratory Regulation of immune dysfunction in cancer, Department of Oncology, University of Lausanne, CH-1066 Epalinges, Switzerland; gregory.verdeil@unil.ch; 2Aix Marseille University, Centre National de la Recherche Scientifique (CNRS) UMR7280, Institut National de la Santé et de la Recherche Médicale (INSERM) U1104, Centre Immunologie Marseille-Luminy (CIML), Parc Scientifique de Luminy, Case 906, 13288 Marseille CEDEX 09, France; lawrence@ciml.univ-mrs.fr (T.L.); amsverhulst@gmail.com (A.-M.S.-V.); 3Centre for Inflammation Biology and Cancer Immunology, School of Immunology & Microbial Sciences, Faculty of Life Sciences and Medicine, King’s College London, London SE1 1UL, UK

**Keywords:** inflammation, tumor-associated macrophages, adoptive T cell therapy, immune suppression, STAT transcription factors

## Abstract

Oncogene-induced STAT3-activation is central to tumor progression by promoting cancer cell expression of pro-angiogenic and immunosuppressive factors. STAT3 is also activated in infiltrating immune cells including tumor-associated macrophages (TAM) amplifying immune suppression. Consequently, STAT3 is considered as a target for cancer therapy. However, its interplay with other STAT-family members or transcription factors such as NF-κB has to be considered in light of their concerted regulation of immune-related genes. Here, we discuss new attempts at re-educating immune suppressive tumor-associated macrophages towards a CD8 T cell supporting profile, with an emphasis on the role of STAT transcription factors on TAM functional programs. Recent clinical trials using JAK/STAT inhibitors highlighted the negative effects of these molecules on the maintenance and function of effector/memory T cells. Concerted regulation of STAT3 and STAT5 activation in CD8 T effector and memory cells has been shown to impact their tumor-specific responses including intra-tumor accumulation, long-term survival, cytotoxic activity and resistance toward tumor-derived immune suppression. Interestingly, as an escape mechanism, melanoma cells were reported to impede STAT5 nuclear translocation in both CD8 T cells and NK cells. Ours and others results will be discussed in the perspective of new developments in engineered T cell-based adoptive therapies to treat cancer patients.

## 1. Introduction

Inflammation is now considered as a hallmark of cancer [1] and the inflammatory context in many cancers is strongly linked to poor prognosis and resistance to therapy. Activating mutations in oncogenic RAS/BRAF/MEK pathways trigger a tumor-intrinsic inflammatory network with the concerted regulation of master transcription factors (TFs) including STAT3 [2], NF-κB and AP-1 [3,4] which in turn trigger the expression of cytokines, including IL-6, IL-1, IL-10, TNF and VEGF [5]. The presence of cytokines is a major regulator of immune cell differentiation/function and is a crucial factor to consider for immunotherapy protocols. Many of these cytokines signal through the stimulation of STAT TFs. In this review, we will comment on the role of STAT TFs (i) for the recruitment and function of tumor-associated macrophages (TAM) and (ii) for the regulation of T cell functions. As STAT TFs are key players in the regulation of functions of these immune cells, their manipulation can have a beneficial or detrimental effect on the anti-tumor response depending on the targeted cell type.

## 2. Tumor-Induced Inflammation Drives Accumulation of Tumor-Infiltrating Myeloid Cells

Cytokine receptor-induced signaling sustains and amplifies the activation of STAT3, NF-κB and AP-1 in a positive amplification loop, fueling tumor-associated inflammation. As such, a core inflammation-related gene set regulated by STAT3, NF-κB and AP-1 has recently been proposed as an “inflammatory index” in breast cancer cell lines and patient samples [4]. Importantly, in correlation with this inflammatory index, this study reported a concerted regulation of (i) non-inflammatory genes related to angiogenesis, metastasis, and cell proliferation; (ii) tumor genome instability; and (iii) heterogeneity of the tumor microenvironment (TME), including the recruited immune cells. Remarkably, these co-regulated characteristics were found across several cancer types driven by distinct oncogenes [4].

Tumor-derived cytokines have been linked to the accumulation of immune-suppressive myeloid cells including both myeloid-derived suppressor cells (MDSCs; IMCs) and tumor-associated macrophages (TAM). In both human and mouse melanomas, IMCs have an important role in malignant progression and evasion from anti-tumor immunity that is linked to the suppression of T cell responses [6,7,8,9]. While increased TAM accumulation has been ascribed to a poor prognosis in established tumors [10], this notion should be refined given the extreme phenotypic and functional heterogeneity of these cells during tumor growth.

Cancer-related inflammation (i) promotes the recruitment of monocyte-derived cells into the tumor bed and (ii) acts systemically as shown by the dysregulated transcriptomic signature of circulating monocytes in breast cancer patients as compared to healthy controls [11]. Numerous therapeutic attempts (reviewed in [12]) to block cytokine-induced monocyte recruitment are under clinical trials using either blocking mAbs (anti-CCR2; anti-CSF1R) or small inhibitors for downstream cytokine receptor signaling (receptor tyrosine kinase inhibitor, Pexidartinib; CSF1R inhibitor, PLX3397).

Once recruited to the tumor, monocyte progenitors (CD11b^+^ SiglecF^−^ Ly-6G^−^Ly6C^+^ F4/80^−^ CD169^−^ MHC-II^−^) undergo a multistep differentiation program [13,14], passing through an immature stage (Ly6C^+^ F4/80^−^ CD169^int^ MHC-II^hi^) before reaching a mature state (Ly6C^−^ F4/80^+^ CD169^hi^ MHC-II^hi or low^). Recently, the heterogeneity of TAM in several cancer types has been emphasized by single-cell RNA-sequencing paired with mass cytometry, including lung [15], kidney [16], breast and endometrial cancers [11], as well as in mouse sarcoma [17]. In the case of renal cell carcinoma, 17 TAM subsets have been characterized [16], most of them expressing CD169 while being discriminated by expression of CD163, CD204 and CD206. Of note, when compared to steady-state tissue-resident macrophages or monocytes, TAM exhibited peculiar gene expression profiles [11,16], highlighting a tumor-induced dysregulation.

There is compelling evidence for high levels of phenotypic plasticity in macrophages, exhibiting differential functional programs depending on their surrounding microenvironment. In response to microbial stimuli pro-inflammatory macrophages—often referred to as of “M1-type” - express cytokines supporting T cell activation. However, in malignancies, alternatively activated macrophages - referred as of “M2-type-secrete cytokines that sustain tumor growth and exert immune suppressive functions. Therefore, differential signal transduction pathways downstream of cell surface sensors define gene expression programs underlying anti- or pro- tumor functions of macrophages [18,19]. As such, IFNγ stimulates STAT1/STAT2 and IRF1/8, which can further promote pro-inflammatory macrophages. While GM-CSF/STAT5 activates a pro-inflammatory signature in monocytes [20], it rather induces a unique reparative program in macrophages after sterile renal injury [21]. Tumor-induced inflammation involving the IL-6/gp130/STAT3 [22] and ERK5/STAT3 [23] axes was shown to drive a pro-tumor transcriptomic program. Additionally, we recently showed that ovarian cancer cells reprogram macrophages towards an IL-4/AKT/STAT6-mediated tumor-promoting phenotype through increased cholesterol efflux from the TAM membrane [24]. Moreover, IL-6 synergizes with IL-4 in the programming of human monocyte-derived macrophages through the concerted activation of STAT6 and STAT3 DNA-binding activities [25]. Given the spectrum of STAT3-regulated genes in TAM encoding pro-tumor and immune suppressive mediators (Table 1), STAT3-modulators are currently being developed to dampen the cancer supportive functions of TAM, as reported in this issue by Rébé and Ghiringhelli [26]. As such, the specific targeting of a highly immune suppressive TAM (CD11b+ CD163+) subset by liposome-encapsulated STAT3-inhibitors showed some success in reprogramming TAM towards a pro-inflammatory profile [27].

## 3. Re-Educating TAM to Restore Anti-Tumor T Cell Functions.

Extensive studies are being conducted to reprogram pro-tumoral TAM towards an inflammatory T cell supporting profile (recently reviewed in [38]). Mature TAM exert both trophic functions—through the promotion of angiogenesis and tissue remodeling—and immune regulatory functions. Here we will focus on the immune aspects of TAM functions, even though these two activities might be closely inter-connected. Indeed, prolonged interactions between stromal TAM (mainly CD163^+^ CD206^+^) and CD8 tumor-infiltrating lymphocytes (TILs) observed by dynamic imaging microscopy, are limiting CD8 T cell motility and their consecutive access to both human lung squamous cell carcinomas and mouse MMTV-PyMT tumors [39].

Recent parallel analyses of TAM and TILs from cancer patients have greatly expanded our knowledge on the reciprocal regulation of these cell lineages. Paired CyTOF-based analyses of CD8 TILs and TAM in human renal cell carcinomas [16] showed some correlation between exhausted CD8 TILs, CD4 regulatory T cells and a few peculiar TAM subpopulations (either CD169^-^ CD163^-^ CD68^hi^ CD38^hi^ CD204^+^; or CD169^+^ CD163^+^ CD68^hi^ CD38^hi^ CD204^+^ CD206^+^). These molecular data had been further correlated with clinical features, with the result that patients with the former TIL/TAM subsets showed increased cancer progression.

We recently reported [13] that a subset of mature CD163^+^ TAM present in mouse melanomas, exhibit transcripts related to T cell immune suppression. Interestingly, a high proportion of CD163^+^ macrophages expressing phospho-STAT3 have been observed in human skin tumors [23]. Targeted-depletion of this minor CD163^+^ TAM subset enhanced melanoma-infiltration by CD8 T cells and promoted CD8 T cell-mediated tumor regression in mice [13]; this was also accompanied by the recruitment of fresh monocytes and immature macrophages with an immune stimulatory phenotype (Figure 1).

Cancer treatments have recently benefited from immune checkpoint therapies (ICT) targeting inhibitory signaling pathways in T lymphocytes (PD-1/PD-L1 and CTLA-4). Despite remarkable success in a subset of patients (objective responses ~20–30% for monotherapy and ~30–40% for combined therapy), primary resistance is observed in more than 50% of the patients and a subset of initial responders can later develop acquired resistance [40]. Analysis of pre-treatment tumor biopsy samples has revealed that patients with a pre-existing local CD8 T cell infiltrate (T cell inflamed) were more likely to show a clinical response to anti- PD-1/PD-L1 [41]. Furthermore, PD-L1-expression on dendritic cells and macrophages but not on cancer cells has been shown to shape the response to the PD-L1/PD-1 blockade [42,43].

Given the aforementioned reciprocal regulation of TAM and TILs, one should expect that ICT-based therapies could conjointly unleash tumor-specific T cells and reprogram TAM functions. Several groups exploring immune cells dynamics in vivo before and after ICT recently addressed this issue. In carcinoma-bearing mice, immune-PET monitoring showed that ICT induced an important infiltration of CD8 T cells within the tumor in responders as compared to non-responders; the responders displayed CD11b^+^ TAM with a “M1-type” transcriptomic signature [44]. scRNA-seq and longitudinal CyTOF analyses also revealed ICT-induced dynamic changes in both TAM and TILs present in induced mouse sarcomas [17]. This study identified an ICT-induced decrease of both CD4 regulatory T cells and exhausted CD8 T cells. Meanwhile, conventional CD4 T cells and tumor-specific CD8 T cells displayed an activated profile as did recruited NK cells, with all these effector cells exhibiting an enhanced NF-κB and IFNγ-driven gene signatures. ICT consequently conferred a remarkable dynamic remodeling on intra-tumor monocytes and macrophages. Distinct TAM populations expressing IFNγ-induced iNOS, while down-regulating IL-4-responsive CD206 proteins, were enriched upon ICT Pathway analyses, which showed ICT-driven enrichment of IFNγ and NF-κB signaling as well as glucose metabolism. Concurrently, alternative “M2-type” TAM clusters with a MerTK^hi^ CD206^+^ CCL2^+^ CD274^+^ IL4Rα^+^ phenotype were reduced by approximately one-third. Kinetic CyTOF analyses further highlighted an important crossroads between days 7 and 9—a time window where the TME drives the TAM behavior towards a pro-inflammatory/T cell supporting or an anti-inflammatory/immune suppressive program. The TME agent mediating monocyte/macrophage reprogramming was ascribed to be IFNγ secreted by ICT-unleashed tumor-specific T cells. Importantly, T cell produced IFNγ not only engages its receptor signaling on surrounding TAM but also on tumor cells themselves. As such, STAT1-activation was reported to sensitize hepatocarcinomas to T cell mediated killing [45] (see Figure 1). Therefore, the therapeutic use of STAT/JAK-modulators should be carefully evaluated for their specific targeting of STAT3 activity while preserving IFNγ-induced STAT1/2 signaling, which appears to be beneficial for anti-tumor immune responses.

In the large fraction of patients who are non-responsive to ICT other therapeutic approaches must be proposed to reinvigorate anti-tumor responses. Providing an infusion of autologous tumor-specific CD8 T cells manipulated to express an effector (IFNγ-proficient) program could be a way to fight cancer and induce TAM reprogramming. This will be addressed in the next section.

## 4. STAT5 versus STAT3 in Adaptive CD8 T Cell Responses to Cancer

Cytokines are major regulators of T cell differentiation/function and are crucial factors to consider for immunotherapy protocols. Many of these cytokines signal through the stimulation of STAT3 and/or STAT5. This makes these two TFs key players in the regulation of functions of T cells. Both STAT3 and STAT5 activation can have a beneficial or detrimental effect on the anti-tumor response depending on the T cell type targeted. In this part, we will focus on the T cell-intrinsic role of these TFs and on the possibility to manipulate them for improving CD8 T cell anti-tumor responses.

### 4.1. STAT5 in the Adaptive T Cell Responses

The STAT5 pathway is predominantly activated in T cells by members of the γc cytokine family (IL-2, -7, -9 and -15) (reviewed in [46]). These cytokines signal through receptors containing the γc (CD132) subunit and lead to the activation of the JAK3 tyrosine kinase. In addition, activation of JAK1 occurs through its interaction with the IL-2Rβ (CD122) (for IL-2 and IL-15), but also, with the IL-4Rα, IL-7Rα and IL-9Rα units. For IL-2, high-affinity binding to its receptor requires the association of an additional subunit (IL-2Rα or CD25) to the IL-2Rβ and the γc components. IL-15 signaling is further peculiar in that it requires the association “in-trans” of the IL-15Rα component with the IL-2Rβ and the γc components [47]. The specificity of signaling by these cytokines is thus partially explained by the recruited STAT proteins, but also by the differential expression of the relevant receptors. STAT5-activating cytokines have a general role in the maintenance and expansion of T cell subsets. IL-2 has a central role in the expansion phase of T cells following their primary antigenic stimulation [47,48]. The roles of IL-7 or IL-15 are rather associated with the maintenance of naïve or memory T cells [47,49,50].

In adaptive immunity, except for a population of CD4 T cells with a regulatory function (Treg), IL-2Rα (CD25) is not expressed at a significant level at the surface of naïve CD4 and CD8 T cells. Although the two other chains of the IL-2R are expressed at low basal levels on naïve T cells, IL-2 fails to activate these T cells, consistent with the report that only Tregs respond promptly to in vivo IL-2 exposure [51]. Thus, antigen stimulation of the TCR/CD3 complex is a pre-requisite for the secondary signaling by the IL-2R. TCR/CD3 initiated signals, which do not involve STAT activation, lead to transient expression of genes, including *il2ra*. IL-2R mediated signaling through STAT5 activation further increases CD25 expression at the cell surface and leads to the stabilization of a panel of genes initially induced by the TCR [52,53]. We and others reported that TCR-initiated signaling is influenced by the avidity of TCR-peptide/MHC interaction, which impacts strength and duration of TCR engagement [54,55,56]. Strong TCR/CD3 stimulation causes production of IL-2, leading to an autocrine effect on IL-2R signaling, which is not induced by low avidity Ags. In this latter case, exogenous IL-2 or expression of a constitutive-active STAT5 protein [52,53] can serve as a substitute for the lack of the IL-2/IL-2R amplification loop. Interestingly, a study evaluating the IL-2R/JAK-regulated phospho-proteome in CD8 CTL revealed a dominance of proteins that control mRNA stability and components of the protein translational machinery leading to an accumulation of cytokines and effector molecules, as well as proteins that support metabolic processes essential for cell “fitness” and important oxygen-sensing components [57]. Of note, active STAT5 does not substitute for all IL-2 induced signaling events that also involve MAP kinase- and phospho-inositol 3-kinase-dependent pathways [58]. Altogether, for CD8 T cells, depending on their state of activation and on the dose of cytokine provided, IL-2 may amplify effector function and proliferation and induce terminal differentiation or activation-induced cell death [47,48].

The synergistic action of TCR-induced TFs and STAT5 in CD8 T cells mirrors the cooperation between STAT5 and (i) TCR-induced GATA3 in CD4 Th-2 cells to control the accessibility of *Il4* gene locus [59]; (ii) Tbet in Th-1/Tc-1 for the regulation of the *Ifng* locus [60,61]; and (iii) BCL6 in B lymphocytes for the generation of memory B cells [62]. Additionally, STAT5 activation was shown to promote GM-CSF [63] and IL-9 [64], producing T cells and to be a prerequisite for Foxp3-expressing Tregs [65,66]. By contrast, STAT5 is a negative regulator of Th-17 [67] and T-Fh [68] by competing with STAT3 and BCL6, respectively. Altogether, STAT5 appears to control secondary decisions in adaptive immunity (see Table 2).

#### Role of STAT5 in the Generation of Memory CD8 T Cells and Maintenance of Effector Functions

Several STAT5A mutants have been shown to confer cytokine-independent growth to Ba/F3 cells [72]. While a STAT5A S710F single-mutant displayed a strong constitutive activity sufficient to complement STAT5A^null^ bone marrow progenitors, a STAT5A double-mutant H298R/S710F was not able to complement STAT5A^null^ cells [80]. Further studies highlighted that the H298R mutation in the DNA-binding domain renders this protein inactive unless it can form heterodimers with endogenous wild-type STAT5 (see Figure 2). Therefore, the constitutive activity of STAT5 H298R/S710F is modest and limited by the bioavailability of wild-type STAT5 to pair with. By contrast, the single STAT5 S710F mutant exerts a strong constitutive activity in both CD8 T cells [81] and Ba/F3 cells [82].

Retroviral expression of STAT5A H298R/S710F (hereafter referred to as STAT5ca) in in vitro activated CD8 T cells led to the generation and maintenance of long-lived CD8 T effector cells upon their adoptive transfer [83]. Transcriptomic analyses of STAT5ca-expressing CD8 T cells highlighted a role for STAT5ca in the stabilization of a broad Tc-1 gene expression program initiated by TCR stimulation [60] (see Table 2, Figure 2). This observation is in agreement with the reported chromatin interactions of STAT5 in super-enhancers to activate IL-2 highly inducible genes [71].

Of note, the in vivo maintenance of STAT5ca-expressing CD8 T cells remains under the control of γc-cytokines (IL-7, IL-15) and TCR tickling by self MHC class I [81]; these properties again point towards a moderate and controlled activity of this double-mutant. Accordingly, Kaech’s group also reported that STAT5ca promoted memory CD8 T cells [49] that did not display any sign of transformation. However, Moriggl and colleagues recently demonstrated that high expression of S710F gain-of-function mutated STAT5A induced PTLC-nos (Peripheral T cell leukemia and lymphoma—not otherwise specified) cells when expressed during T cell development in transgenic mice [84].

Mice expressing a constitutively active STAT5Bca (H298R/S715F) transgene in the lymphoid lineage have been shown to present a selective expansion of memory-like CD8 T cells. Their analysis further suggested that moderate STAT5B activation underlies both IL-7/IL-15-dependent homeostatic proliferation of naive and memory CD8 T cells and IL-2-dependent development of CD4 CD25^+^ Tregs [85]. When expressed in the B cell lineage in mouse models, STAT5Bca (H298R/S715F) induces B cell acute lymphoblastic leukemia thanks to cooperative molecular events targeting JAK1 activity, tumor-suppressor genes, and pre-BCR signaling [86]. Indeed, mutated STAT5Bca was shown to antagonize preBCR-initiated TFs (NF-κB, IKAROS) for binding to B cell specific super enhancers [87]. Finally, mice which expressed a transgene, i.e., a human gain-of-function mutation of STAT5B (hSTAT5B N642H) identified in leukemic patients, developed lymphomas from multiple T cell subsets [88]. The recent crystal structure of hSTAT5B N642H highlighted important conformational changes in correlation to its resistance to dephosphorylation [89]. Overall, strong STAT5B hyperactivity appeared to trigger B or T lymphomas when express during lymphoid cell development and to directly influence disease aggressiveness and therapeutic resistance [87,89].

### 4.2. Role of STAT3 in T Cells

Several cytokines/cytokine receptors trigger STAT3 signaling in T cells, including IL-6-type cytokines (IL-6 and Oncostatin M), IL-10, IL-17, IL-21 and IL-27. Among the STAT3-activating cytokines, IL-21 is peculiar: While its receptor contains the γc (CD132) subunit, signals downstream of IL-21R converge on STAT3 and to a lesser extent on STAT1, rather than on STAT5 activation [46]. Intriguingly, IL-21 was shown to promote in vitro proliferation of CD8 T naïve and memory cells in synergy with STAT5-inducing cytokines IL-7/IL-15 [90]. Indeed, Leonard’s group reported that IL-2 vs. IL-21 cytokine stimulation of pre-activated T cells induces STAT5 and STAT3 binding, respectively, to incompletely overlapping super enhancers [71]. Understanding the fine-tuning of cytokine-modulated STAT3/5 activities will require further effort as reviewed in [46]: This tight regulation relies on the T cell context, including cell lineage, differentiation state, and cytokine milieu, triggering synergistic/antagonistic STAT-mediated transcriptional regulation. Thus, the amplitude of STAT3 activation differs from one T cell subset to another in correlation with differential expression of the corresponding T cytokine receptor [91]. Interestingly, chronic TCR engagement was recently shown to enhance the sensitivity to IL-6/STAT3 signals [92], indicating again a cross-talk between TCR-induced and STAT TFs.

The function of STAT3 in various T cell subtypes is summarized in Table 2, but further details are described in this issue by both Rébé and Ghiringhelli [26] and Logotheti and Putzer [93].

The overall effect of STAT3 stimulation in T cells is generally correlated with a poor cytotoxic/increased regulatory response and as such, is associated with a defective anti-tumor immune response [94]. In the next part, we will focus on the role of STAT3 in the maintenance and formation of memory CD8 T cells.

#### 4.2.1. Role of STAT3 in the Maintenance and Memory Formation of CD8 T Cells

Several studies have highlighted the role of IL-21 and STAT3 activation in T cell memory regulation. Using LCMV chronic infection in mice, Kaech’s group showed that IL-10 and/or IL-21-induced STAT3 was required for the differentiation of memory CD8 T cells with a preserved replicative potential [95]. Indeed, in the absence of STAT3, memory precursor cells were replaced by T effector-like cells that failed to undergo homeostatic proliferation or to protect against secondary infection. Interestingly, STAT3 appeared to act both by induction of “pro-memory” TFs, such as Eomes and BCL6, and by dampening inflammation-driven TFs, including STAT4 and Tbet.

A similar role for STAT3 in human T cells [96] was reported in a cohort of patients suffering from autosomal-dominant hyper-IgE syndrome, which is caused by dominant-negative STAT3 mutations. These patients have increased numbers of naïve T cells but fewer central memory CD4^+^ and CD8 memory T cells, leading to an increased susceptibility to EBV infections. This was associated with a defect of these naïve T cells to express the TFs important for memory formation.

Accordingly, IL-21 repressed the cytolytic effector program in tumor-specific CD8 T cells while preserving their replicative potential [97].

#### 4.2.2. Role of STAT3 in the CD8 T Cell-Mediated Anti-Tumor Responses

STAT3-activating cytokines such as IL-6 and IL-10 are found in abundance within the tumor microenvironment. It was clearly shown that STAT3 inactivation in tumor-specific CD8 T cells increased their potential for tumor elimination after adoptive transfer [98,99]. Altogether, these studies suggest that a general inhibition of STAT3 in T cells would be a beneficial treatment for increasing tumor control. However, contradicting data exist in the literature [100]. In mice bearing transplanted tumors, a combination of anti-OX40 mAb and TGFβ/Smad inhibitors improved the CD8 T cell mediated tumor rejection. Use of OX40-Cre STAT3^fl/fl^ mice further demonstrated that in absence of STAT3 in OX40-expressing T cells, the response to the treatment was lost [100]. These authors did not prove formally that the deletion of STAT3 in CD8 T effector cells only was mediating the same effect; but their data suggest that depending on the T cell subset or the immunotherapy used, inhibiting STAT3 could become detrimental and lead to poor survival of T cells and loss of the benefit of the treatment.

The reported discrepancies for the function of STAT3 in CD8 T cell mediated anti-tumor responses may rely on distinct roles of this TF depending on the T cell differentiation state (effector versus memory), the combined immunotherapy regimen and the STAT3-inducing cytokines in the TME. All these parameters may contribute to differential STAT3-activation thresholds triggering distinct T cell responses, a hypothesis that needs to be tested in refined analyses.

## 5. Adoptive CD8 T Cell Therapy and CAR-T Cell Generation for Cancer Immunotherapy

### 5.1. Increasing the Frequency of Tumor-Specific CD8 T Cells

As mentioned earlier in Section 3, beside tremendous achievements for patients with various types of cancers, immunotherapies aimed at blocking inhibiting receptors still failed for more than 50% of cancer patients. This could be the consequence of the paucity of tumor-specific CD8 T cells, an obstacle that could be circumvented by infusions of in vitro amplified tumor-specific T cells. Indeed, Adoptive Cell Transfer (ACT) of amplified TILs and Chimeric Antigenic Receptor (CAR) treatments are based on the use of the patient’s T cells [101]. Historically, T cells used for ACT were obtained from tumor pieces/biopsies, amplified in vitro and transferred back into the patient when a sufficient number was reached (up to 10^11^ cells) [102]. The progress in gene engineering removed many hurdles to generate cancer specific T cells. CAR-T cells are gene-engineered products obtained from the expression in T cells of a construct encoding a domain of a single-chain variable antibody fragment recognizing a tumor antigen, fused to intracellular signaling motifs inducing T cell activation upon recognition of the target cells [103]. T cells from the blood are used for this therapy, allowing an increased potential to generate these cells on a large scale. CD19-directed CAR-T cells have shown impressive results in the treatment of patients with B-cell lymphoma [104]. However, evidence for the use of CAR-T cells against solid tumors is still sparse. The main limitation for application being the identification of Ags expressed exclusively on solid tumors and not on the non-transformed cell counterparts. Defining tumor-specific Ags for solid cancers is still a challenge [105,106]. Such an attempt was reported using mesothelin-specific CAR-T cells to treat patients bearing pancreatic ductal adenocarcinoma [107]. Of note, antibody-targeted chimeric T cells display receptors (i) of higher avidity for tumor-Ags than regular TCRs; (ii) which are not MHC restricted and thereby, are not impacted by tumor editing in the Ag processing/presentation pathway, (iii) interacting only with Ags expressed at the tumor cell surface.

Despite some clinical success using ACT or CAR-T cells, much effort is being made to improve T cell survival and resistance to immunosuppression in the frame of these therapies. Given the ability of IL-2 to expand CD8 T cells and to maintain their cytotoxic function, infusion of recombinant IL-2 in cancer patients was the first reported effective immunotherapy for human cancer [108]. However, IL-2 concomitantly acts on Tregs that suppress anti-tumor responses, and systemic IL-2 injections had severe adverse effects including vascular leak syndrome and pulmonary edema due to IL-2Rα expression on endothelial cells [109]. This has led to the development of (IL-2/anti-IL-2 mAb) complexes [110]: depending on the mAb the binding of the resulting complex is favored on IL-2Rβ/γc-expressing cells such as the memory CD8 T and NK cells, but prevented on IL-2Rα^+^ Tregs and endothelial cells. Such approaches enhance the cytokine half-life in vivo and were shown to induce a massive (>100-fold) expansion of CD8 T cells in vivo [110,111] thereby boosting anti-tumor immune responses in mice and humans [112].

IL-21 is also considered to expand less differentiated CD8 T memory stem cells with both proliferative and multipotent potential [113] to be used in ACT.

The manipulation of cytokine-induced STAT-signaling in tumor-specific CD8 T lymphocytes may be another solution to potentiate adoptive T cell cancer therapies.

### 5.2. On CD8 T cell Intrinsic Modifications For adoptive T Cell Therapies

#### 5.2.1. Promoting Effector Memory CD8 T Cells Through STAT5

Another hurdle for ACT is the persistence of the infused T cells, their capacity to home in tumor-invaded peripheral organs and to resist the local immune suppression. As such, CD19-directed CAR-T cells had limited capacities to control the growth of CD19^+^ murine melanomas in relation with their scarcity in the tumor [114]. To infiltrate non hematologic solid tumors, T cells must acquire an effector phenotype: this includes the simultaneous loss of CD62L (L-selectin) and CCR7 expression, and acquisition of tissue-specific homing molecules such as adhesion molecules and chemokines/chemokine receptors [115]. Given the multiplicity of the molecular partners involved in T cell egress and migration, it is illusory to manipulate these targets, one by one or in combination. However, manipulating TFs known to be activated during T effector cell differentiation may have broader effects. We brought this proof of concept by expression of STAT5ca during the in vitro differentiation of CD8 T cells, which modified their in vivo migration upon ACT with increased infiltration of both non-lymphoid tissues in healthy mice and melanomas in tumor-bearing mice [60]. At the transcriptomic and protein levels, coincident down-regulation of CD62L and up-regulation of CCR2, CCR5 and CXCR3 was induced by STAT5ca as compared to standard CD8 T effector cells. Altogether, STAT5ca-CD8 T cells acquired a long-lived effector memory phenotype in correlation with a concerted STAT5-mediated regulation of *Tbx21* and *Eomes* genes (Table 2 and [60]). Coupled to their enhanced effector functions—Ag-driven cytotoxity and IFNγ production—these engineered STAT5ca-CD8 T cells mediated efficient melanoma rejection as compared with conventional CD8 T effector cells [83]. Importantly, this STAT5ca-induced reprogramming applies to both polyclonal and monoclonal (TCR transgenic) CD8 T cells, after primary or secondary stimulation [83]: these characteristics are of particular importance for translation to cancer patients’ CD8 T cells. Finally, STAT5ca-expression in tumor-specific T cells had better therapeutic potential than the potentiation of standard CD8 T effector cells with combined infusions of IL-2 complexes, the effect of which disappeared at the end of treatment [83].

#### 5.2.2. Promoting Central Memory CD8 T Cells Through STAT3

Recent studies have highlighted the importance of a memory-like CD8 T cell population to unleash an anti-tumor response by anti-PD-1 treatment [116,117]. In these studies, a small subset of CD8 TILs expressing PD-1 and the transcription factor TCF1 responded to anti-PD-1 immunotherapy by differentiating into highly cytotoxic TILs that mediated long-term tumor control. These studies highlight the importance of a long lasting, less differentiated population to mediate a more efficient control of the cancer.

This point is also valid for ACT, as transfer of expanded T cells that are too differentiated could lead to a failure of the treatment related to a lack of survival of the cells [118]. For that purpose, IL-21, a STAT3 stimulating cytokine, is efficient in driving the differentiation of central memory CAR or TCR engineered CD8 T cells in vitro [119]. Additionally, a recent study showed that CD19-directed CAR-T cells initially activated in vitro with optimal stimulation (anti-CD3/CD28) progressively acquired an “exhausted” phenotype upon transfer in tumor bearing mice. Thus, when retrieved from the tumor 12 days after their transfer, they presented a characteristic high surface expression of inhibitory receptors, diminished production of IFNγ and TNFα, and low cytolytic activity as well as the expression of the high-mobility group (HMG)-box TFs - TOX and TOX2. Expression of the latter TFs was recently described as resulting in commitment of CD8 T cells to an exhausted transcriptional and epigenetic developmental program distinct from that of functional CD8 T effector and T memory cells [120]. To understand further the factors associated with an efficient CAR response, Fraietta et al. [121] performed a transcriptomic profiling of CAR-T cells from complete-responding patients with chronic lymphocytic leukemia compared to non-responders. The gene signature was enriched in memory-related genes, including IL-6/STAT3 signatures, whereas T cells from non-responders upregulated programs involved in effector differentiation, glycolysis, exhaustion and apoptosis. Moreover, highly functional CAR-T cells from patients produced the STAT3-activating cytokine IL-6 and its concentration in serum was correlated with CAR-T cell expansion. These results showed STAT3 signaling is beneficial for an efficient CAR-T cell response [121]. This finding is further validated by the use of a new-generation CD19-CAR that encodes a truncated cytoplasmic domain from IL-2Rβ and a STAT3-binding tyrosine-X-X-glutamine motif. These CAR-T cells showed antigen-dependent activation of the JAK kinase and of the STAT3 and STAT5 TFs signaling pathways, which promoted their proliferation and prevented their terminal differentiation in vitro. The CAR-T cells demonstrated superior in vivo persistence and anti-tumor effects in models of hematologic malignancies as compared with CAR-T cells expressing a CD28 or 4-1BB co-stimulatory domain alone [114]. It should be noted that while a subtle increased STAT3 activation by these newly engineered IL-2Rβ-CAR-T cells was therapeutically beneficial, STAT3 gain-of-function mutants induced multi-organ autoimmunity and large granular lymphocytic leukemia [122].

In conclusion, these studies strengthen the interest of regulating STAT3/STAT5 activities in CD8 T cells to improve adoptive cell therapy: given their tightly balanced contribution to T cell memory fates, increased STAT3 or STAT5 activity is expected to improve the therapeutic success of hematological and non-hematological cancers, respectively. This hypothesis has to be validated in relevant preclinical mouse models.

#### 5.2.3. On the STAT5 versus STAT3 Balance in T Cells

Several groups have provided data showing that the competitive binding of STAT3 and STAT5 dimers to DNA dictates the transcriptional regulation of certain target genes in T cells and thereby either trigger autoimmune disorders or promote tumor rejection (some examples are reported in Table 2). These data highlight the need for both in silico analyses [123] and appropriate cell/animal models to evaluate the outcome of competition between STAT TFs on T cell fates.

## 6. STAT5 and Resistance to Immunosuppression in the Tumor Microenvironment

Chronic stimulation of tumor-specific T cells promotes an exhausted phenotype, sharing strong [124] but not-completely overlapping similarities [125] with virus-induced exhausted CD8 T cells. Both the diversity and the high expression of inhibitory receptors by CD8 TILs (reviewed in [126]) can contribute to ICT resistance. However, combining blocking antibodies towards several inhibitory receptors to improve anti-tumor responses may further increase the adverse effects among which autoimmune attacks [127]. Therefore, manipulating CD8 T cells to resist local tumor-derived immune suppression may be of clinical benefit.

### 6.1. Immunosuppression by PD-1

PD-1 is related to the CD28 superfamily and is expressed on activated T cells, B cells, monocytes and macrophages. Engagement of PD-1 by its ligands (PD-L1 or PD-L2) induces inhibitory signals in T cells [128] through the induced phosphorylation of ITIM/ITSIM motives in its cytoplasmic domain, which leads to recruitment of the SHP-2/SHP-1 phosphatases. This dampens both TCR and CD28 signaling leading to abrogation of the PI3K/AKT and ERK pathways, as shown notably in primary CD4 T cells [129]. Interestingly, SHP-2 recruited by PD-1 was shown to counteract phospho-STAT5 signaling in human Tregs from HCV-infected patients’ livers [130]. It was also demonstrated that PD-1 intrinsically regulates a subpopulation of high PD-1 expressing innate cells (ILC2) involved in helminth immunity (γc dependent proliferation) by inhibiting STAT5 phosphorylation through SHP-1/SHP-2 phosphatases [131]. Similarly, for T cells, the inhibitory effect of PD-1 is observed on populations expressing high levels of surface PD-1 such as exhausted T cells in contrast to acutely activated T cells [132]. However, in a melanoma model, in spite of upregulation of the PD-1/PD-L1 immunosuppressive pathway in the tumor microenvironment, STAT5ca-expressing CD8 T effector cells were found to be resistant to inhibition by PD-1/PD-L1 engagement as measured by their efficient IFNγ production, in contrast to their control counterpart [132]. Thus, the previously reported increased resistance to dephosphorylation of STAT5ca [72] may also contribute to its resistance to the phosphatases recruited to checkpoint inhibitors (Figure 2) [133,134].

### 6.2. Other Immune Suppressive Pathways

The diversity of immune suppressive mechanisms associated with inflammatory melanoma have previously been reported [133], with several involving tumor-derived proteins.

TGFβ1, an immunosuppressive cytokine produced by a number of cancer cells, was shown to inhibit both Th-1/Tc-1 cytokine production and Tc-1/NK cytolytic effector functions through the repression of *Ifng, Gzmb,* and *FasL* gene transcription [135,136]. Interestingly, STAT5ca-expressing T cells maintained efficient induction of both IFNγ and Gzmb in the presence of TGFβ1 [132]. The insensitivity to TGFβ1 may be linked to the STAT5ca-mediated negative regulation of *Tgfbr2* expression as well as its positive effects on *Ifng* and *Gzmb* gene transcription [60]; thus, STAT5ca-gene regulation counterbalances the transcriptional repression exerted by TGFβ1-induced Smad2/3. Indeed, two STAT5-activating cytokines, IL-2 [136] and IL-15 [137], have been shown to counteract TGFβ1-mediated immune suppression.

Of note, while TGFβ1 and IL-6 contribute to *Maf* up-regulation in exhausted CD8 T cells [125], STAT5ca-expressing T cells displayed reduced levels of *Il6st*, *Tgfbr2*, and *Maf* transcripts [60] and maintained a functional antitumor program.

Another tumor-secreted protein, the Dickkopf-related protein 2 (DKK2), was shown to be overexpressed in human colorectal cancers and melanomas, and to promote tumor progression by suppressing the activation of both CD8 T and NK cells [138]. Mechanistically, tumor cells secrete DKK2 that binds its receptor LRP5 inducing interactions between the intracellular domain of LRP5 and STAT5; while STAT5-phosphorylation was not precluded, STAT5 nuclear localization was impeded in cytotoxic cells. Conversely, DKK2 blockade enhanced cytotoxic immune cell activation.

Altogether, maintenance of active STAT5 signaling in cytotoxic cells appears to correlate with efficient anti-tumor immunity [73,83].

## 7. Conclusions

The frequency of tumor-specific CD8 T cells can now be increased in cancer patients through sophisticated personalized medical approaches including in vitro derived and expanded TILs or engineered CAR-T cells. Nevertheless the ability of adoptively transferred T cells to survive in cancer-invaded hosts and to efficiently penetrate into the tumor are dampened by the solid tumor landscape including cancer cells, tumor stroma and immune suppressive myeloid cells. Therefore, enhancing tumor T cell infiltration is one way to improve cancer immunotherapy. This could imply (i) manipulation of the T cells themselves through their transduction with chemokine receptors promoting non-hematopoietic tissue colonization [139,140] or manipulation of TFs that regulate expression of these homing receptors [60]; (ii) depletion of specific TAM subsets that exert trophic and/or immune suppressive functions either by ablation [13] or inactivation using STAT3 inhibitors [27]. The enhanced killing of cancer cells together with the recruitment of fresh monocytes will favor the engagement of the host immune system thanks to the presentation of tumor-derived Ags by professional APCs such as dendritic cells and/or immature TAM. This would result in spreading of tumor-derived epitopes ensuring a polyclonal anti-tumor response that could ultimately prevent antigen escape. Importantly, this favorable loop was observed when using STAT5ca-expressing CD8 T cells in ACT as their efficient capacity to infiltrate and kill inflammatory melanomas is accompanied by a concomitant infiltration of host T cells [83]. Compared to systemic treatment, such as IL-2 infusions, the targeted manipulation of STAT5 activity in tumor-specific CD8 T lymphocytes circumvents the negative stimulation of Tregs, which are also recruited into tumor beds. We here propose to transiently express (i.e., through mRNA transfection for example) the STAT5A H298R/S710F mutant in human CD8 T cells specific for Ags expressed on solid malignancies to be used in ACT: Both their efficient infiltration into tumors and IFNγ production resistant to tumor-derived suppression should help to restore endogenous T cell responses and to reprogram TAM. Although the IFNγ/STAT1 signature induced by T cells has recently been reported as a good prognostic factor for ICT response [141], a close control of this cytokine-induced signaling must be maintained since excess IFNγ can induce death of anti-tumor CD8 T cells at a certain stage of their differentiation/activation [142,143,144].

## Figures and Tables

**Figure 1 cancers-11-01832-f001:**
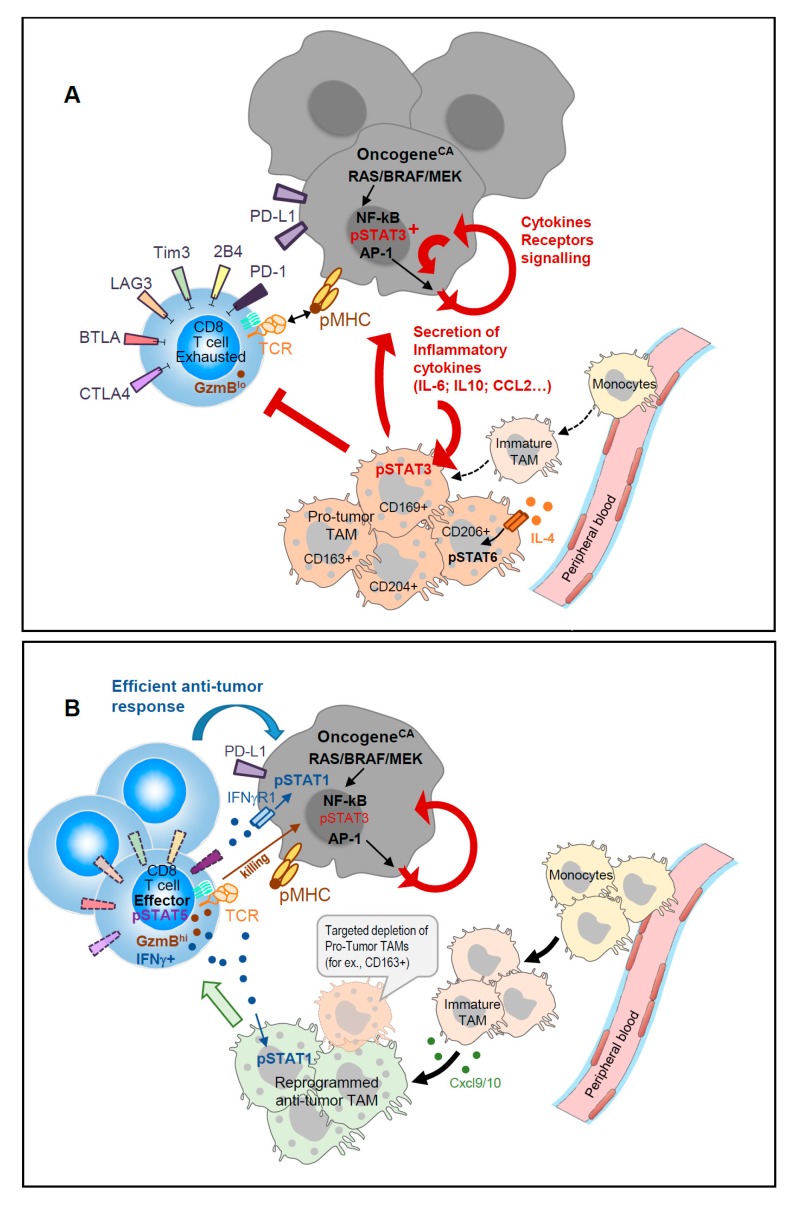
(**A**). Concerted regulation of tumor-induced inflammation promotes accumulation of immune-suppressive TAM. Oncogene activation in tumors induced secretion of inflammatory cytokines that activate STAT3 and promote the accumulation of immune-suppressive TAM; IL-4-derived signals through STAT6 sustain the pro-tumor TAM function that further sustains tumor growth in a positive feedback loop. (**B**). Targeted depletion of specific TAM subsets displaying strong T cell suppressive activity favors the recruitment of monocyte-derived immature TAM that secrete CXCL9/10 and display T cell activating capacities. Reinvigorated CD8 TILs with activated pSTAT5 secrete IFNγ that further maintains TAM anti-tumor functions and sensitizes tumors to T cell killing.

**Figure 2 cancers-11-01832-f002:**
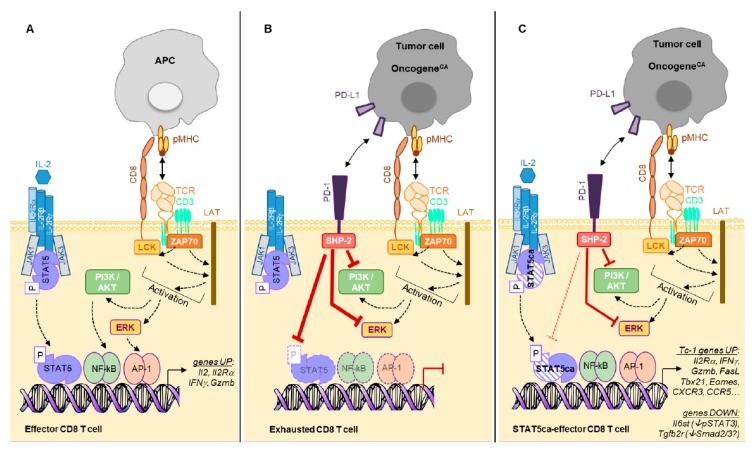
(**A**). Ag presented by APC (Ag-presenting cells) triggers a T cell activation cascade leading to gene transcription including *Il2* and *IL2Rα* genes. Binding of IL-2 to its receptor further amplifies the TCR-initiated gene transcription program. (**B**). Ag expressed on tumor cells mediates chronic TCR engagement on CD8 TILs leading to their “exhaustion”, which is characterized by expression of multiple inhibitory receptors (as shown in Figure 1). For simplicity, we represent PD-1 only that recruits the phosphatase SHP-2 mediating inhibition of ERK and PI3K/AKT pathways as well as dephosphorylation of STAT5. (**C**). Expression of STAT5ca (H298R/S710F, here represented by dashed symbols as compared to the wild type (WT) protein) in CD8 T cells not only recapitulates the IL-2-mediated TCR-initiated gene transcription, but also stabilizes this functional program. This leads to a sustained Tc-1 program reminiscent of effector memory cells. Of note, while being PD-1^hi^ due to the chronic TCR engagement by their cognate Ag, STAT5ca-expressing T cells remain functional, as the S710F substitution reduces the SHP-2-mediated dephosphorylation. Additionally, STAT5ca represses the expression of *il6st* and *Tgfb2r* genes, rendering these cells insensitive to IL-6/STAT3 and TGFβ1/Smad signaling.

**Table 1 cancers-11-01832-t001:** STAT3–regulated genes in tumor-associated macrophages (TAM).

Target Genes	STAT3 Input	Cancer Type	TAM Phenotype
Cytokines/Cytokine receptors			
*Il10 **	positive	m-melanoma	CD11b+ [28]
		m-PDAC	CD68+, IL-10Ra+ [29]
*Il23a **	positive	m-melanoma	CD11b+ CD11c− [30]
*Il10ra, Il4ra*	positive	m-PDAC	CD68+, IL-10Ra+ [29]
*Tgfb1 **	positive	m-melanoma	CD11b+ CD11c− [30]
*Il12a*	negative	m-melanoma	CD11b+ [28]; CD11b+ CD11c− [30]
*Ifng*	negative	m-melanoma	CD11b+ [28]
Chemokines/Chemokine Receptors			
*Ccl5, Cxcl9-10-11*	Negative	m-melanoma	CD11b+ [28]
*Cxcl2, Cxcl12*	positive	m-melanoma	BMDM+Tumor conditioned media [31]
			MDSC: CD11b+ GR1+ CD11c− [22]
Scavenger receptors/Endocytosis			
*Mrc1 (CD206)*	positive	m-breast	CD11b+ Ly-6C^lo^ F4/80^hi^ CD24^lo^ MHC-II^lo^ [32]
*CD163*	positive	h-gastric	CD163+ CD209a+ [33]
		h-SCC	ERK5+ CD163+ [23]
		m-PDAC	CD68+, IL-10Ra+ [29]
*Cd209a*	positive	m-PDAC	CD68+, IL-10Ra+ [29]
Immune suppression			
*Arg1*	positive	m-PDAC; h-PDAC	CD68+, IL-10Ra+ [29]; blood CD14+ [34]
*Cox2*	positive	m-melanoma	BMDM + Tumor conditioned media [31]
*Ido1*	positive	m-liver metastasis	liver-MDSC: CD11b+ Ly-6C^int/hi^ Ly-6G^+^ [35]
*Pdl1 (CD274)*	positive	h- & m-glioma	h-CD68+; m-CD11b+ CD115+ [36]
		h-breast	CD163+ [37]
		m-liver metastasis	liver-MDSC: CD11b+ Ly-6C^int/hi^ Ly-6G^+^ [35]
Extra-cellular Matrix/Angiogenesis			
*Mmp2*	positive	m-melanoma	BMDM + Tumor conditioned media [31]
*Vegf*	positive	m-melanoma	CD11b+ [28]; MDSC: CD11b+ GR1+ CD11c- [22]
*Cathepsin (B, L)*	positive	m-PDAC	CD68+, IL-10Ra+ [29]
Cell cycle/TFs			
*Ccnd1*	positive	m-melanoma	BMDM + Tumor conditioned media [31]
*ATF6, sXBP1*	positive	m-PDAC	CD68+, IL-10Ra+ [29]

* Immune suppressive cytokines.

**Table 2 cancers-11-01832-t002:** Concerted gene regulation by STAT3 and STAT5 in helper and cytotoxic lymphocytes.

STAT3 and STAT5-Regulated Genes in T and NK cells				
Target Genes	STAT3 Input	Cell Subsets	STAT5 Input	Cell Subsets
Cytokines/Cytokine receptors				
*Il17*	positive	hCD4 Th-17 [69]		
*Il10*	positive	mCD4 Th-2 [70]		
*Il2ra (CD25)*	positive	mCD4 Th-21 [71]	positive	mCD4 Th-1 [71], mCD8 [60], mNK [72]
*Il7ra (CD127)*			negative	mCD8 [60]
*Il6st*			negative	mCD8 [60]
*Tgfb2r*			negative	mCD8 [60]
Homing				
*CCR7*			negative	mCD8 [60]
*Sell (CD62L)*			negative	mCD8 [60]
TFs				
*Prdm1 (Blimp)*	positive	mCD4 Th-2 [70]		
*Eomes*			positive	mCD8 [60]
*Tbx21 (Tbet)*			positive	mCD8 [60]
Cytotoxicity				
*Gzmb*			positive	mCD8 [60], mNK [73]
*Prf1*			positive	mCD8 [60], mNK [73]
*Ifng*			positive	mCD8 [60], mNK [73]
*Fasl*			positive	mCD8 [60]
Cell cycle/Cell survival				
*Myc*	positive	hT lymphoma [74]	positive	mCD8 [60]
*Pim1*	positive	hT lymphoma [74]	positive	mCD8 [60,75]
*Bcl2*	positive	hT lymphoma [74]	positive	mCD8 [49,60]; NK [73]
**Competitive gene regulation by STAT3 and STAT5 in T cells ***				
**Target Genes**	**STAT3 Input**	**STAT5 Input**	**T Cell Subsets**	
*Il17*	positive	competitor	mCD4 Th-17 [67]	
*Il9*	competitor	positive	mCD4 Th-9 [64,76]	
*Bcl6*	positive	competitor	mCD4 Tfh; mCD4 Th-1 [68,77]	
*Socs3*	positive	positive	mCD4 Th-17/Treg balance [78]	

* Mechanisms underlying competitive gene regulation by STAT3 and STAT5 are reviewed in [79].

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
