# Peer review of "Targeting STAT3 and STAT5 in Tumor-Associated Immune Cells to Improve Immunotherapy"

_cancers, 2019, doi:10.3390/cancers11121832_

Round 1

Reviewer 1 Report

In this review article Verdeil et al discuss the roles of STAT3 and STAT5 in certain aspects of tumor immunity and their potential use as targets for immune therapy. They focus their discussion on two immune cell subsets, specifically tumor associated macrophages and CD8 T cells. The text is well-written, sufficiently cited and the concepts and ideas are thoroughly discussed. This review has enough details for the specialized scientists but is also simple enough for the non-specialists, which is how a review article should be. 

The authors start in section 2 by elaborating on how cancer induced inflammation may generate suppressive TAMs and how these cells may affect subsequent immune responses in favor of the tumor. They then present evidence for the role of STAT3 and STAT5 in these cells, and how TAMs can be manipulated in order to be become "immunogenic". However there doesn't seem to be a direct role for STAT3 and STAT5 in this regard. Although this is apparent in one of the two last paragraphs of section 2, the authors might want to discuss more whether direct targeting of STAT3 and/or in TAMs is possible and its potential outcomes.  Since the authors primarily focus on CD8 T cells, it would be preferable if this is shown in the section titles and abstract. Thus, instead of referring to T cells, the authors should refer to CD8 T cells. Otherwise, as a reader one expects to find a literature review on the role of STAT3 and STAT5 in CD4 and gamma delta T cells, both of which are important for cancer immunity. In particular, the need for CD4 T cells for CD8 memory and the direct role of cytotoxic gamma delta T cells or the indirect pathogenic role of IL-17-producing gamma delta T cells.  When the authors discuss STAT5 and immunosuppression it will benefit if they added a paragraph or so on regulatory T cells (Tregs), since they are critically dependent on STAT5 and have been shown to be important in many cancer models.  A few minor points and semantics: 1) preferably use "T cells" instead of "T-cells", 2) line 29: use " others' " instead of " other's " since it is plural, 3) line 58: consider not using "so-called", 4) line 61: remove "was", 5) line 84: are M2 macrophages the suppressive TAMs? it's not very clear, 6) line 87: IFNg signaling is via both STAT1 and STAT2, 7) line 95: add "of" after "spectrum", 8) line 138: use "dynamic" instead of "dynamics", 9) line 139: use "decrease" instead of "decreased", 10) line 145: add "and" before "glycolytic signaling"; also here the term signaling with glycolytic is not correct; consider glycolysis or glucose metabolism, 11) line 154: use "has" instead of "have", 12) line 42: use "STAT" instead of "STATs" 13) be consistent with STAT nomenclature; sometimes it is all caps, sometimes it is small letters. When referring to protein, all letters should be capital, eg STAT5A and STAT4B instead of STATb, Stat5b or STAT5a  

Author Response

We thank the reviewers and the academic editor for their comments. We hope and trust that our detailed answers to the reviewers' comments have clarified and improved this manuscript.

-      All highlighted (yellow) paragraphs have been reformulated: §4.1 (lines 200-210; 235-237);§4.2.1 (lines 304-315); §6.1 and 6.2. We have also added two sentences in the concluding remarks (lines 490-496) to explain better our perspectives for ACT in cancer patients.

-      We have introduced a paragraph (lines 220-227) describing the role of STAT5 in Tregs and other T cell subsets.

-      As requested, we have referred to CD8 T cells in section titles and abstract.

-      We have mentioned in lines 68-69 some strategies to deplete monocyte-derived tumor-infiltrating cells.

-      We have proposed at the end of section 2 a strategy to modulate STAT3 activity in immune suppressive TAM (lines 99-101).

-      We have included all the minor corrections proposed by the reviewer and we carefully homogenized the nomenclature. Accordingly, STATa is now STATA; STATb is now STAT5B; to avoid any misunderstanding, we have changed STAT5CA for STAT5ca.

The manuscript has been edited by an american immunologist.

Given the nomenclature revison that introduced numerous text substitutions, I here enclosed a corrected version of the manuscript in which only important text modifications are highlighted (yellow).

Reviewer 2 Report

This review by Verdeil and colleagues nicely discusses the literature on the roles of STAT3 and STAT5 in T-cell responses, in the complex functions of tumor-infiltrating immune cells, and the implications for cancer immunotherapy strategies. Overall, it is presented well and is a nice contribution to the field. This reviewer requests some minor points to be addressed before publication:

The section 4.1 on STAT5 in adaptive T cell responses would benefit from briefly expanding on the role of STAT5 in different T cell subsets. Particularly, the sentence in lines 191-192 is quite general and could be expanded to describe the positive role of STAT5 on differentiation of CD8 T cells, Th1, Th2, Th9, Th-GM and Tregs, and the negative regulation of Th17 and Tfh. Also, the authors should elude to the fact that CD25 is a target gene of STAT5 in paragraph 3 of this section (starting line 205).

The section beginning on line 228 should include a better explanation of the reasoning behind the use of the H299R/S710F STAT5a double mutant. How does the loss of DNA-binding and heterodimer formation-dependent activity make this a useful or physiological mutant to study the role of STAT5 in T cells? Is it because the H299R mutation dampens the highly proliferative, oncogenic phenotype driven by the S710F mutation, such that you are left with a persistently active but non-transforming variant that allows you to model more physiological activities of STAT5 in T-cells? A summary sentence at the end of this section would also help the readers to conclude on what the mutational studies described in this section reveal about STAT5 function in effector memory T cells. Can the authors also discuss other studies examining STAT5 functions in memory and effector T cells that used other systems?

Also in this section, the authors refer to the mutant S710F but on lines 242 and 246 it is referred to as S711F. Also, on line 264 for Stat5b it is listed as S711F, but residue 711 in Stat5b is not a serine. This should be corrected.

Line 224: The ‘It’ at the beginning of the sentence – does this refer to IL-2-activated STAT5? Perhaps the authors can be more specific in what is meant here.

Lines 66-69: It is understood that the clinical trials are reviewed in another cited review, but it would be helpful to the reader if the authors listed a couple of examples of such mAbs and small molecule inhibitors. Also, should the reference on line 69 be [12] instead of [125]?

A final concluding sentence at the end of the paper would be helpful on what the authors conclude on the potential benefits and feasibility of using the STAT5CA mutant in the therapeutic strategies discussed.

There are several spelling and grammar mistakes throughout the manuscript and English editing is recommended. For example (but not limited to), the sentence on lines 143-145 is hard to understand without appropriate grammar. Line 250: the word ‘cells’ is missing after ‘CD8 T’. Line 61: delete ‘was’. Line 77: by expression ‘of’. Line 95: Given the spectrum ‘of’, etc.

Line 228: This section should start as 4.1.1.

Line 201 and 204 should both refer to ‘gene expression profile’.

Lines 437-438 and 450: Should be written ‘PD-L1 or PD-L2’ instead of ‘PDL-1 or PDL-2’.

Author Response

We thank the reviewers and the academic editor for their comments. We hope and trust that our detailed answers to the reviewers' comments have clarified and improved this manuscript.

-      All highlighted (yellow) paragraphs have been reformulated: §4.1 (lines 200-210; 235-237);§4.2.1 (lines 304-315); §6.1 and 6.2. We have also added two sentences in the concluding remarks (lines 490-496) to explain better our perspectives for ACT in cancer patients.

-      We have introduced a paragraph (lines 220-227) describing the role of STAT5 in Tregs and other T cell subsets.

-      As requested, we have better described the molecular regulation of the STA5ca mutant (lines 235-237).

-      We have corrected the positions of Serine mutations in both STAT5A and STAT5B thorough the manuscript.

-      We have mentioned in lines 68-69 some strategies to deplete monocyte-derived tumor-infiltrating cells.

-      We have included all the minor corrections proposed by the reviewer and we have carefully homogenized the nomenclature. Accordingly, STATa is now STATA; STATb is now STAT5B; to avoid any misunderstanding, we have changed STAT5CA for STAT5ca; we checked the writing PD-L1/PD-L2.

The manuscript has been edited by an american immunologist.

Given the nomenclature revision that introduced numerous text substitutions, I here enclosed a corrected version of the manuscript in which only important text modifications are highlighted (yellow).
